# Novel Repair Procedure for CFRP Components Instead of EOL

**DOI:** 10.3390/ma14112711

**Published:** 2021-05-21

**Authors:** David Rabe, Philippa Ruth Christine Böhnke, Iris Kruppke, Eric Häntzsche, Chokri Cherif

**Affiliations:** Institute of Textile Machinery and High Performance Material Technology (ITM), Helmholtzstr. 5, 01069 Dresden, Germany; philippa.boehnke@tu-dresden.de (P.R.C.B.); iris.kruppke@tu-dresden.de (I.K.); eric.haentzsche@tu-dresden.de (E.H.); chokri.cherif@tu-dresden.de (C.C.)

**Keywords:** CFRP, repair, textile, patch, thermoset, novel method, CF, UV-based depolymerization

## Abstract

Today, numerous carbon fiber (CF) reinforced plastic (CFRP) components are in continuous usage under harsh environmental conditions. New components often replace damaged structural parts in safety-critical applications. In addition to this, there is also no effective repair method to initially restore the mechanics of these structures using dry fiber material. The high costs of CFRP components are not in proportion to their lifetime. The research project IGF-19946 BR “CFRP-Repair” addresses this specific challenge. By using an oxide semiconductor that is activated by ultraviolet (UV) irradiation, the thermoset matrix can be depolymerized and thus locally removed from the damaged CFRP component. Afterward, the harmed fibers can be physically removed from the laminate in this certain area. A load-adjusted tailored fiber reinforcement patch is subsequently applied and consolidated by local thermoset re-infiltrating. Using this procedure, the structure can be locally repaired with new CF. As a result, repaired CFRP structures can be obtained with reduced mechanics and an approximately original surface. This article gives an insight into the developed repair procedure of CFRP components in an innovative and more efficient way than the state-of-the-art.

## 1. Introduction

Carbon fiber reinforced plastics (CFRP) offer a high lightweight design potential due to their specifically high mechanical characteristics, which is why they are widely used in many industries, e.g., automotive, wind energy, civil engineering, sports and leisure [1,2,3,4,5]. CFRP components usually have high production costs and restricted recyclability and, above all, poor repair opportunities [1,2,6]. Previous methods for repairing CFRP are mainly based on shape cutting (e.g., milling) the damaged composite area, the comprising fibers and the matrix. The repair site of the CFRP component prepared in this way is then mostly filled up with pre-impregnated layers, so-called prepregs [7,8,9,10]. There are several approaches to repair CFRP structures, e.g., the scarf method or doubler method [7,11,12,13]. These initial repair processes are always associated with high manual effort and manufacturing expenses and often significantly reduce the composite strength of the repaired component or cause extra weight. In most cases, complete parts or components have to be replaced [14,15]. The early end of life (EOL) leads to a decrease in profitability of the use of CFRP and a decrease in its lightweight design potential. Furthermore, this significantly reduces the reliability of CFRP production, which is, in any case, highly energy-consuming.

Recent research efforts have been focused on the development of a repair process based on oxide semiconductor-supported matrix degradation. In the affected area, the matrix can be removed by an ultraviolet (UV) irradiation-initialized degradation mechanism, so-called depolymerization [16]. This process can be used for two scenarios:(1)‘Matrix-repair’ can be used if the damage of the composite is narrowed to the matrix and the fibers are all intact. The matrix is then removed completely from the affected area, leaving the dry fibers that remain in it. After a surface activation, the repair site can be thermoset-reinfiltrated, and the composite can be restored without cutting any fiber.(2)‘Patch-repair’ can be used if both components of the composite are damaged, i.e., the fibers and the matrix. The matrix is removed initially in the affected area as in the first scenario. Subsequently, defective CF can be removed. The fibers are cut manually and staggered. A textile patch is fitted in the prepared gradation, and the area is re-infiltrated to restore the composite.

## 2. Materials and Methods

### 2.1. Materials

Within the project, the CFRP samples were used to develop the repair process. The CFRP material was made of a stack of four two-layered biaxial reinforced non-crimp fabrics (NCF) with a symmetric layering ((0/90)_2_)_sym_. The NCF was based on Toray T700SC 50C 12K, 800 tex CF roving, provided by Saertex GmbH and Co. KG (Saerbeck, Germany). The areal weight was 314 g/m^2^ per NCF ply. The reinforcement material was processed in a resin transfer molding (RTM) process to square plates with a fiber volume content of approximately 60%. The cavity height and the following sample thickness was 1.5 mm, preheating was done at 60 °C and an RTM was performed at a pressure ascending from 1 to 6 bar. After the production of the plates, the material was tempered for 15 h at 80 °C. The gravimetric ratio between the used hardener RIMH137 and the reference epoxy resin RIMR135 was 30 to 100, both supplied by Hexion GmbH, Duisburg, Germany. The ‘reference’ was represented by the virgin CFRP-material without damage.

In addition, the effect of two additives on the mechanical properties was tested. The additive BYK-C 8013 is a polymeric coupling agent for increasing the mechanical strength of radically curing systems, and BYK-P 9920 is an additive for improving fiber wetting in CFRP. Both additives can help to improve the connection between the existing components and the patch material. Therefore, resin systems and additives were combined in different ways, as shown in Table 1. HS1 is equal to the resin system of the ‘reference’ CFRP sample plates. The difference is the speed of cross-linking. The hardener used for re-infiltrating shows an even faster coupling and hardening than the RimH137. HS2 represents the combination of the reference resin system with a low mixing ratio and HS5 with a high mixing ratio of the two additives BYK-C8013 and BYK-P9920. HS3 is the resin system ER0051/EC0052 from Leuna-Harze (Leuna, Germany). This resin system matches well with the reference due to its equal viscosity and hardening time and therefore similar mechanical properties. This resin system is combined with a low ratio of both of the additives in HS6.

Different patch and matrix materials were used for the repair by of a 1.0 mm gap a CNC milling through the complete laminate thickness pre-damaged samples (Section 2.2.2). All patches were based on the same T700SC 50C 12K CF roving as in the NCF ply. One aim was to reproduce the areal weight and reinforcement structure of the original textile reinforcement of the CFRP component with different textile process technologies:CNC-cutting and stacking of the UD material (UD): The UD patches were CNC-cut-outs from a roll of UD material with 151 g/m^2^ areal weight. The CF is fixed by an adhesive thread grid on the back side. After CNC-cutting eight single plies, they were stacked and thermo-bonded with each other (Figure 1a).Tailored fiber placement (TFP): The CF was embroidered onto a thermostable film using Grilon KE85, 166 dtex fusible adhesive thread and thus fixed. Four patch-structures with two layers (0/90°) were produced (Figure 1b). In a downstream process, the patch was fixed by heating above 85 °C through the melting of the fusible adhesive yarn so that the embroidery base could be easily removed.Multilayer weft knitting (MLG): The four biaxial reinforced repair-patches were knitted in one piece and subsequently divided into four structures. The stacked patch-structure can be seen in Figure 1c. In the edge areas, the stitch thread was thermo-bonded to prevent the mesh thread from unraveling.

In addition to the use of different textile patches, the repair resin system was varied to get an insight as to whether it is possible to repair a CFRP component with a non-origin resin system or, even better, the mechanical properties.

### 2.2. Methods

#### 2.2.1. Matrix-Repair

As briefly described in the first section of this is the state-of-the-art scenario for repairing production faults of composites, e.g., voids, porosity or delamination, all defects are only affecting the matrix and not the fiber structure. These defects can occur during the manufacturing, e.g., in the RTM-process. If there are bigger areas with trapped air, normally the parts have to be rejected from the production. To repair such defects, the area for matrix removal is determined by a scanning image processing method, e.g., using ultrasonic waves or eddy currents. In the determined damaged area, a semiconductor (Cerium dioxide: CeO_2_) is applied onto the surface of the CFRP. The polymer chains of the semiconductor are stimulated and split (depolymerization) due to the radical formation initialized by UV irradiation. This may lead to a chain reaction, as long as the energy input is constant. The matrix is decomposed by activating the semiconductor due to UV irradiation. The process products are carbon dioxide and water, and the CF remains intact. Further details concerning the matrix dissolution are the focus of earlier research activities, as described in [16]. The oxide-radical matrix removal process causes a discoloration that appears during the preparation of the repair site. It occurs marginally in the boundary area of the treatment site, where the radical depolymerization does not proceed due to the absence of HLO. It seems obvious that the UV irradiation accelerates optically perceptible aging of the thermoset matrix. In subsequent tensile tests, the structural failure occurred in the treated center region of all observed specimens but not in the discolored edge region. It can be assumed that the discoloration in the edge region is not a primary contributor to the failure and that its influence on the structural mechanics is negligibly small. Any matrix degradation intermediates are removed in the following steps for cleaning and resizing. During the matrix removal process, the original sizing from the CF is removed as well. Before repairing the area by re-infiltrating, the area has to be cleaned and the sizing has to be reapplied to the exposed fibers. The so-called cleaning and resizing are done by a plasma torch with a precursor feed. The sizing (EP 871, Michelman Inc., Cincinnati, OH, USA) activates the surface of the fibers within the plasmatic atmosphere of the torch. The deposition of sizing was set to 16 × 10^−6^ L/min at a moving speed of the plasma torch of 20 mm/min. To get an insight of the effect of the surface activation and resizing, three sample series were prepared. The cleaning and resizing were done in different intensities by the variation of the repetitions:0×—cleaned by plasma torch and surface-activated,1×—cleaned, surface-activated and coated in a single cycle by plasma torch,2×—cleaned and surface-activated and coated in a double cycle by plasma torch.

Within this study, the matrix of CFRP plates with the dimensions 250 mm × 150 mm × 1.5 mm has been removed, followed by cleaning and resizing two sides from the middle of the specimen (marked with a yellow dashed line) within a length of 80 mm (yellow marked area: 150 mm × 80 mm). Each test series consists of ten specimens (25 mm × 250 mm) cut from two CFRP plates (Figure 2).

#### 2.2.2. Patch-Repair

The second scenario for repairing composites briefly introduced in the first section is the repair with the help of a textile patch. Breakages or fine cracks in the composite structure coupled with broken fibers are defects that can be fixed with a patch. The inter load transfer of the reinforcement structure is destroyed. The repair of such defects can be realized with the developed method of local matrix removal from the composite. Therefore, the procedure is equal to the ‘Matrix-repair’ (Section 2.2.1) at the start. The repair area is determined, and the matrix is decomposed out of the composite [16]. The defect fibers remain in a textile condition. To restore the load transfer, the fibers are cut staggered to get an overlap between the textile ends of the repair patch and the unhinged fibers of the composite to be repaired per load bearing layer (Figure 3). By overlapping the patch with the textile fibers ends, the load transmission can be improved by fiber–fiber-friction and an increase of the specific surface for the matrix-coupling in the repaired component.

After removing matrix and the harmed fibers in a staggered scheme, the repair area is surface-activated and resized comparable to the ‘matrix-repair’. The prepared repair area is filled up to the removed fiber mass using a customized repair patch. These textile patches are prepared by three different textile fabric formation procedures (Section 2.1) with an overlap of 20 mm per 0°-layer. The upper layer of the patch has a total size of 160 mm × 150 mm, with the smallest layer on the opposite side 40 mm × 150 mm. Belonging to the patch structure, the fibers are placed in one-piece (UD) or layer-by-layer (TFP, MLG). A metal pressure piece like a counterpart to the metal tool is placed on top of the repair area (Figure 4). After the placement of the patch and the metal counterpart, the whole area is sealed by a vacuum film and tacky tape. The vacuum is applied, the CFRP plates on the metal tool are preheated to 50 °C, and the chosen repair resin system (Section 2.1) is infused using a vacuum-assisted resin infusion (VARI) technique. After the re-infiltration is finished, the plates are put in a hydraulic press to get back the former surface quality. After curing the resin system (40 min), the sample is cooled down and unpacked from the vacuum bagging.

Within the study, CFRP plates (250 mm × 150 mm) have been used to develop and show the repair procedure. Therefore, in the middle of the plate (Figure 5, yellow dashed line), damage is realized by CNC milling of a 1.0 mm gap through the complete laminate thickness. In this way, the damaged samples do not have any residual load bearing capacity, and a good comparison to the undamaged ‘reference’ is possible. The CFRP plates prepared in this way are repaired by the ‘patch-repair’. The series that was prepared and tested on ten specimens (25 mm × 250 mm) from two CFRP plates can be seen in Figure 5. The patch length on the front (patch)-side of the sample is 160 mm.

#### 2.2.3. Tensile Testing (DIN EN ISO 527-4)

The CFRP specimens (25 mm × 250 mm) of the test series ‘reference’, ‘matrix-repair’ and ‘patch-repair’ were tensile tested until full rupture using a Zwick/Roell Z100 tensile tester (Zwick/Roell, Ulm, Germany). All tensile tests were performed according to the DIN EN ISO 527-4 standard [17], where the gauge length was 100 mm, the crosshead speed was set to 2 mm/min and the clamping pressure of the hydraulic clamping was set to 50 bar. The specimens repaired with a ‘patch-repair’ were prepared with shorter cap-strips (40 mm instead of 50 mm) than those prescribed in the testing standard because of the length of the patch. Using a 50 mm cap-strip would have resulted in jamming on the patch area.

#### 2.2.4. Digital Image Correlation (DIC)

During the tensile testing, the straining behavior of the tensile test specimens was recorded by a DSLR camera Eos 7d (Canon, Krefeld, Germany). The surface was marked with a speckle pattern. Based on the captured raw video data, the local strain was calculated by the software GOM Correlate (GOM, Braunschweig, Germany).

#### 2.2.5. Light Microscopy

The repaired samples were cut into smaller sections and embedded into epoxy resin. After this, the samples were tempered and polished. Images from the cross-sections of the samples were taken using the microscope AXIOImager.M1m from Zeiss (Oberkochen, Germany).

## 3. Results

### 3.1. Investigation of the Influence of the UV-Radiation-Treatment, Resizing and Re-Infiltrating (Matrix-Repair)

The results of the tensile test (DIN EN ISO 527-4) for the ‘matrix-repaired’ specimens can be seen in Figure 6. The diagram depicts maximum force F_MAX_ and the elongations ε at F_MAX_ depending on the chosen repair method. As already mentioned in Section 2.2.1, three different series (0×, 1×, 2×) have been characterized, and the results are compared to the ‘reference’. Samples repaired with the ‘matrix-repair’ method show a lowered maximum force F_MAX_. About 66.7% of the maximum force of the ‘reference’ CFRP can be reached. This corresponds to a loss of 33.3% of the maximum force by the UV-irradiation, cleaning, surface activation and resizing and re-infiltrating. With regard to the elongation ε at maximum force, the same results can be seen, with about 70% maximum elongation remaining from ‘reference’. The standard deviation of the ‘matrix-repair’ series values rises in comparison to the ‘reference’. The difference between the series with different preparation (intensity by repetition) is only marginal. The repeated surface activation with an increasing number of repetitions and thus the amount of sizing do not change the maximum achievable force or maximum elongation.

### 3.2. Investigation of the ‘Patch-Repair’ with Different Patches Made with Different Textile Procedures

Figure 7 depicts the results of the tensile test (DIN EN ISO 527-4) performed with ‘patch-repair’ samples. Three different series of damaged samples have been repaired with different patch materials: UD, TFP, MLG (Section 1 and Section 2.1) and the results are compared to the ‘reference’. All patch-materials show almost similar maximum forces F_Max_. Concerning the maximum elongation, the samples repaired with UD and MLG are almost similar, and the samples repaired with TFP were already breaking at 0.5% strain. The ‘patch-repair’ samples show about 51–55% of F_MAX_ and elongation ε at F_MAX_ compared to the reference samples. The standard deviations of the readings (F_MAX_, ε) are in the same range as the reference.

Figure 8 shows three microscopic images of a repaired CFRP area—from the edge to the center of the sample (1–3). It can be seen that the composite area completely replaced with the UD-patch (3) was well impregnated and only a small number of pores are visible. The first and second images depict a higher number of dark areas due to residuals from the UV-induced matrix removal process or pores resulting from these residuals, respectively.

Samples repaired with UD material were examined closer with DIC during the tensile test. Figure 9 shows the calculated strain using GOM Correlate software on one sample. The local stress concentrations just before CFRP breakage from the front and the back side of the sample can be observed. The front side shows the area with the applied patch, and the back side shows the remaining composite of the repaired CFRP plate. The picture shows different inhomogeneous strain fields with strains ε between 0% and 1.5% along the measuring length of 130 mm. On the front side (patch), the elongation ε alternates in total 0.9% (Figure 7, UD) in equidistant steps over the length of the sample. The steps show the prepared stepped sample and the inserted patch. The backside looks similar, but there is a high elongated area in the middle can be seen, which represents the inserted damage of the CFRP by CNC milling (Section 2.2.2).

### 3.3. Investigation of the Repair with Different Resin Materials

Figure 10 shows the results of the tensile test (DIN EN ISO 527-4) for the ‘patch-repair’ specimen series repaired with a UD patch but with different resin systems (cf. Table 1). HS1 is equal to the UD series in Figure 7 due to the same standard resin system. The other series have been repaired by different resin systems to determine the influence of the achievable (chemical) bonding. The standard resin system HS1 exhibits the highest force results, but HS3, HS2 and HS5 are quite close.

## 4. Discussion

### 4.1. Matrix-Repair

It was shown in Figure 6 that the ‘matrix-repaired’ samples lost about one-third of the mechanical properties compared to the ‘reference’ representing the original CFRP-component without any damage. The breaking behavior of the three different prepared series (0×, 1×, 2×) is comparable. In comparison to the clean breakage of the ‘reference’ samples, the ‘matrix-repaired’ samples exhibited a rather brittle fracture behavior. The structural failure of all specimens occurred in the treated center area. The reduction of the maximum force F_Max_ (−33%) and the maximum elongation ε (−23%) is probably due to the matrix residues remaining in the cross-section during the matrix’s UV-irradiation based decomposition. These residues obviously cannot be completely removed even by plasma-based cleaning and surface activation. They act as disturbing foreign bodies during re-infiltration that impair resin bonding. These residues thus prevent complete re-infiltration at the filament level (intralaminar). The incomplete re-infiltration results in higher porosity and insufficient fiber-matrix adhesion due to the local disruption of the load transfer. The brittle fracture behavior of the specimens underlines this assumption of the resulting composite mechanics.

### 4.2. Patch-Repair

In Figure 7 and Figure 10, the results of the tensile test of the sample series ‘patch-repair’ are shown. All samples broke early at 51−55% of the breaking force of the ‘reference’, independent of the applied patch material or infused repair resin system. The specimens of all ‘patch-repair’ series showed a comparable fracture behavior. The specimens fractured along the edge region between the patch and the stock bond. The areas marked in yellow in the schematic diagram (Figure 3) should lead to an increase in the fiber–fiber adhesion due to the overlap of the exposed fiber ends with the fibers of the patch and thus cause high breaking forces. However, the assumption could not be confirmed in this test series. The specimens fractured due to interlaminar shear failure of the matrix in the repair areas. An increase of the breaking force by generating a force flow through the overlapped fibers could not be achieved in this stand. The reason for this is assumed to be the incomplete UV irradiation-based matrix removal in the deeper layers of the composite. If the fibers are not completely in textile condition after UV matrix ablation or repair site preparation, it is not possible to increase the force flow. Only up to 55% of the reference breaking force can be achieved.

With the help of the DIC, the local elongations of the patch area could be determined. In all specimens (Figure 9 representing one result of a ‘patch-repair’ sample with a UD patch reinfused with the reference matrix-system HS1), it can be seen that areas of locally varying strain occur along the length of the specimen. Areas of greater strain occurring on both sides of the specimen are marked. These steps show the staggered scheme of the prepared CFRP specimen and those of the patch, respectively, which are interlocked. In the center of the image of the rear side of the specimen, the inserted damage area can be seen centrally. It seems clear that a higher strain occurs in this area than in undamaged areas. For all repaired CFRP specimens, it can be recognized that the monitored area of the as-built bond (non-patch area) exhibits a significantly higher strain (red) than the bond in the patch area (blue-yellow) up to the complete fracture of the specimen. This suggests that there is no uniform stress distribution across the repaired area. A load transfer into the patch is obviously not fully possible. The failure behavior along the shank can be foreseen by analyzing the images even before the fracture. From the area of greatest strain in the center of the specimen’s back (damage), fractures occurred along the shank to the as-built bond by interlaminar shear failure.

### 4.2.1. Influence of Repair Patch

The textile patch type has only a minor influence on the achievable composite’s mechanical properties. All three patch manufacturing processes deliver similar characteristic values. The performance is obviously dependent on other factors, such as the bonding of the patches or the preparation. Using MLG manufactured repair patches, the composite’s mechanical properties of the UD patch can be reproduced. The values achieved with the TFP process were slightly lower. Compared with the reference, these patch structures also achieved approximately 55% of the original CFRP strength. The specimens repaired with the TFP patch structures showed a comparable failure behavior with the specimens repaired with a UD patch, with a brittle fracture behavior in the overlap area. The fibers of the patch did not break. The observed failure behavior is influenced by the remaining residuals from the UV-induced matrix removal process that correlates with a lot of pores and internal defects in the repaired CFRP structure. Only the MLG specimens showed partial fractures in the as-built bond, which resulted in a displacement of the fracture from the overlap or repair area, respectively. The patch bond is thus slightly more stable.

### 4.2.2. Influence of Repair Matrix-System

As a further remarkable factor in the repair of components, various thermoset resin systems were used for the repair. Figure 10 shows the obtained results. By varying the thermoset resin systems HS2–HS6, no further improvement of the breaking force could be achieved compared to the standard resin system (HS1). The achievable specific values using the resin systems HS2, HS 3 and HS 5 are approximately 10% lower than the standard resin system HS1. This means that a later repair with an alternative resin system performed in a repair shop without the knowledge of the origin resin system is possible.

## 5. Conclusions

With the help of the developed repair procedure for CFRP described in the presented study, completely damaged CFRP components up to a thickness of 1.5 mm and no residual capacity could be repaired with a recovered breaking force of up to 55% of the undamaged ‘reference’ composite. This can be achieved by preparation of the repair area on even CFRP plates using a semiconductor oxide activated by UV-irradiation to initialize the local matrix removal (decomposition) and the repair with textile repair patches made with different textile procedures and an adapted thermoset resin infusions process. It can be concluded that the influence of the textile procedure cannot be affirmed in this study. The breakage in the overlap area by interlaminar shear failure identifies significant differences between the investigated patch manufacturing methods anyway.

To reach 100% of the load bearing capacity of an undamaged CFRP component, further research steps have to be made. In terms of the UV-treatment and the following surface activation, there is some potential regarding the loss of one-third of the breaking force without any damage to the fibers. In order to be able to repair CFRP components and structures that are relevant to practice, it must also be possible to repair thicker and complex shaped CFRP components. Therefore, the three-dimensional processing of direct-fit step patches in the exact geometry that maps the repair area needs to be investigated in further research activities.

## Figures and Tables

**Figure 1 materials-14-02711-f001:**
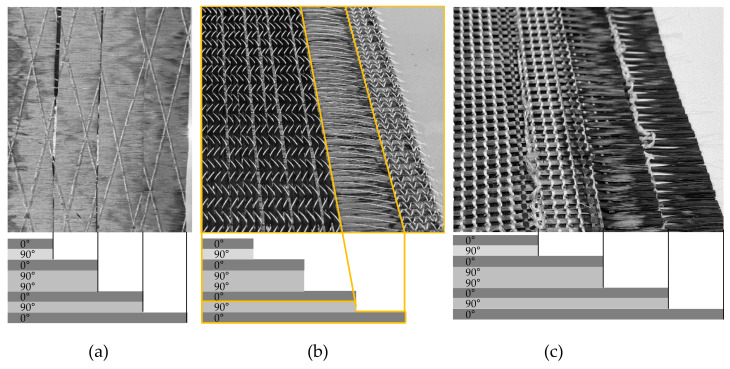
Manufactured repair patches, right half shown, with stacking sequence below: (**a**) full stacked UD patch, (**b**) one part of the full biaxial reinforced two-layered TFP-patch (yellow marked layers of the full patch) and (**c**) stacked, biaxial, reinforced MLG-patch that consist of four double layered patch plies.

**Figure 2 materials-14-02711-f002:**
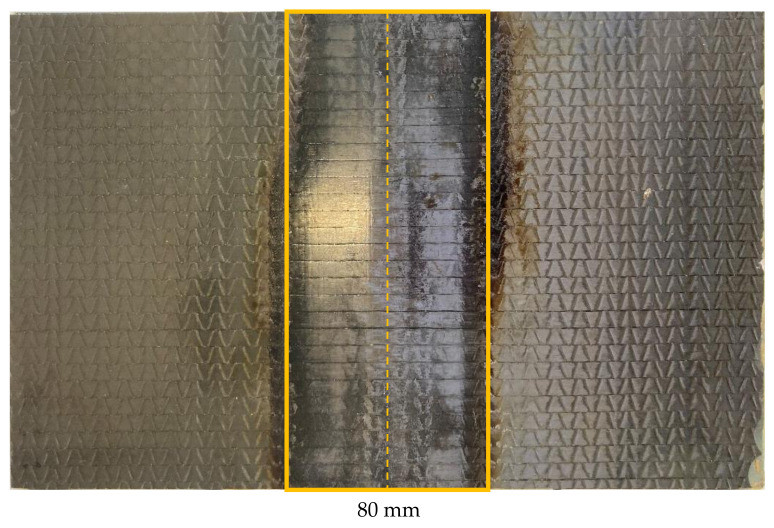
Example for a ‘matrix-repair’ CFRP plate (250 mm × 150 mm) after repair.

**Figure 3 materials-14-02711-f003:**
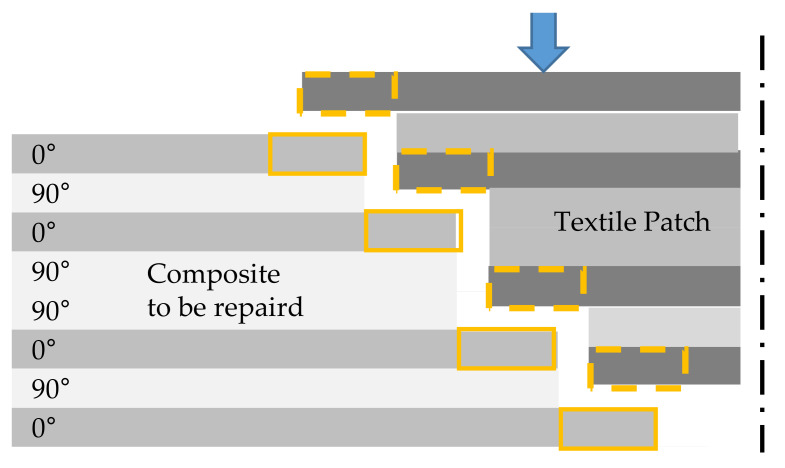
Repair area with the prepared staggered composite with textile fiber ends (yellow), and the textile patch with overlapping fiber ends (yellow dashed line).

**Figure 4 materials-14-02711-f004:**
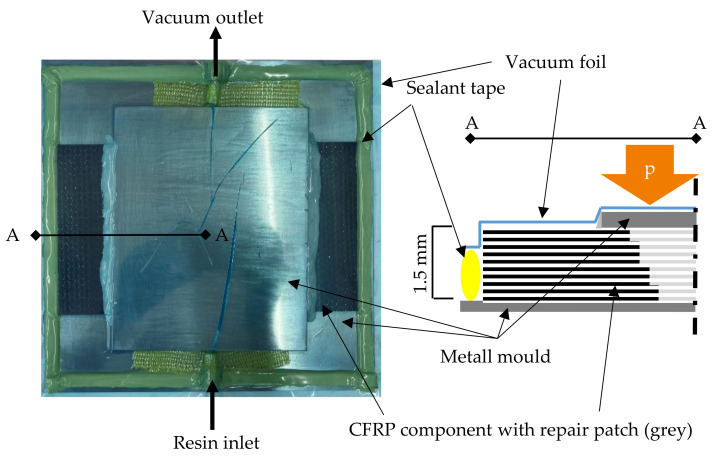
VARI re-infiltration setup for CFRP repair with textile patches.

**Figure 5 materials-14-02711-f005:**
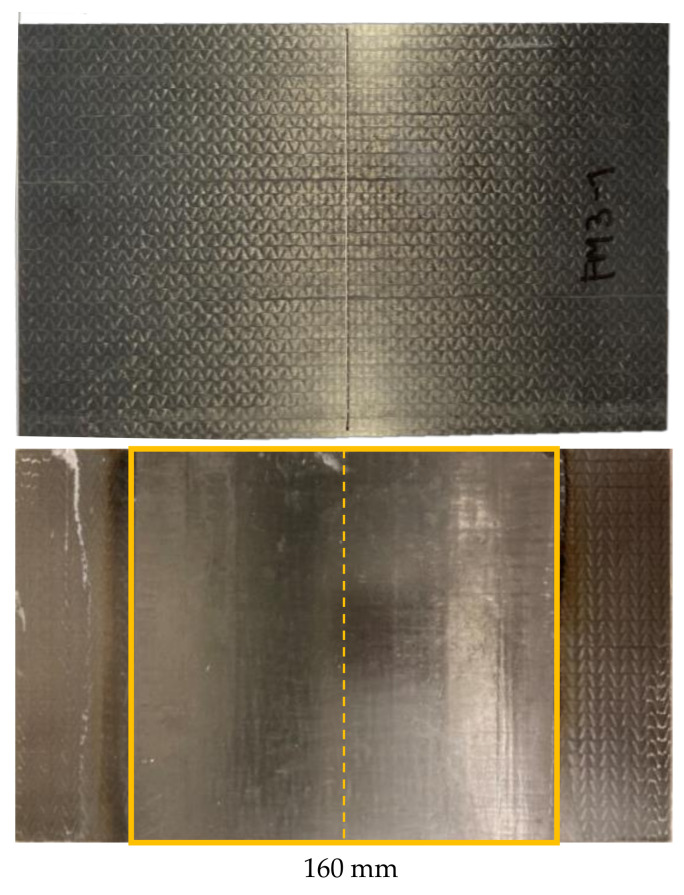
Exemplary damaged CFRP plate (**top**) using CNC milling and a ‘patch-repair’ CFRP plate (250 mm × 150 mm) with a UD patch after re-infiltrating (**bottom**).

**Figure 6 materials-14-02711-f006:**
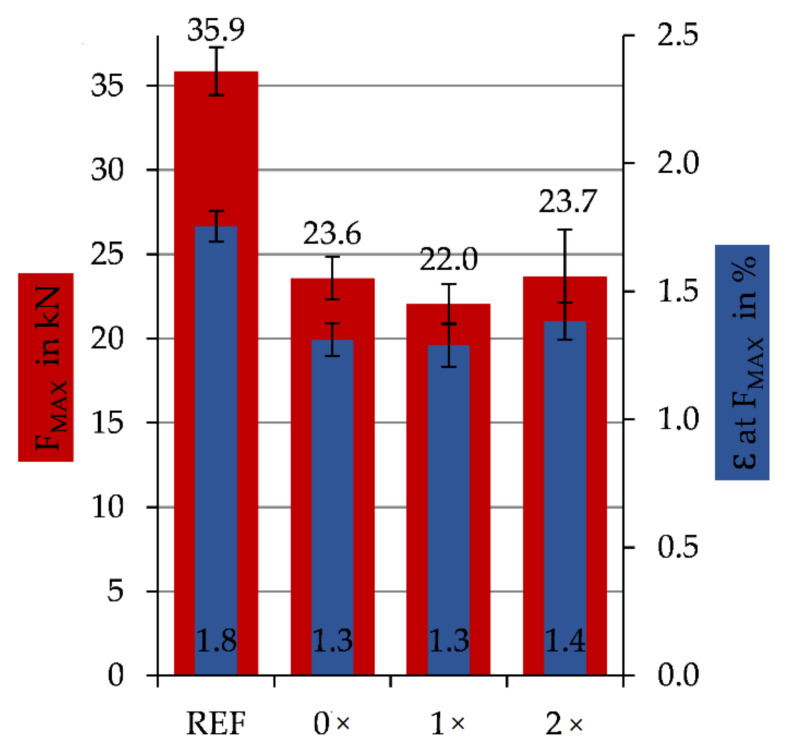
Specific tensile values (DIN EN ISO 527-4) for ‘matrix-repaired’ samples with different cleaning and sizing procedures (0×, 1×, 2×) in comparison to the reference without any defect.

**Figure 7 materials-14-02711-f007:**
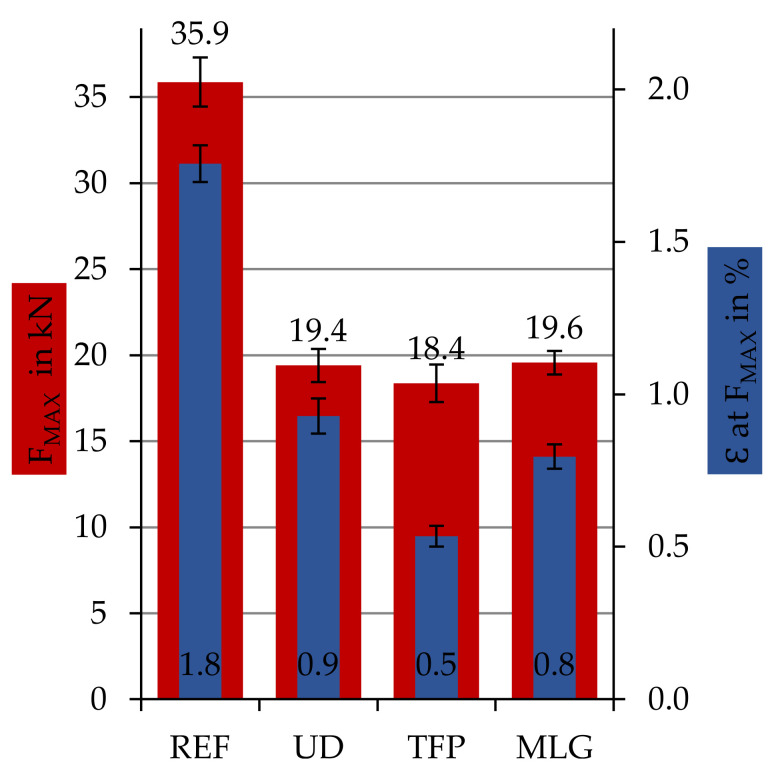
Specific values of damaged samples, repaired with different patch materials—cut UD material (UD), patches manufactured by tailored fiber placement (TFP), patches manufactured by multilayer weft knitting (MLG) compared to the reference samples without any defect.

**Figure 8 materials-14-02711-f008:**
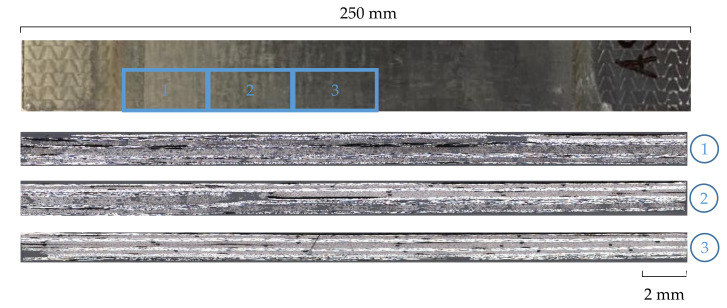
Microscopic cross-sectional views (50×) of different regions of a specimen repaired with a UD patch.

**Figure 9 materials-14-02711-f009:**
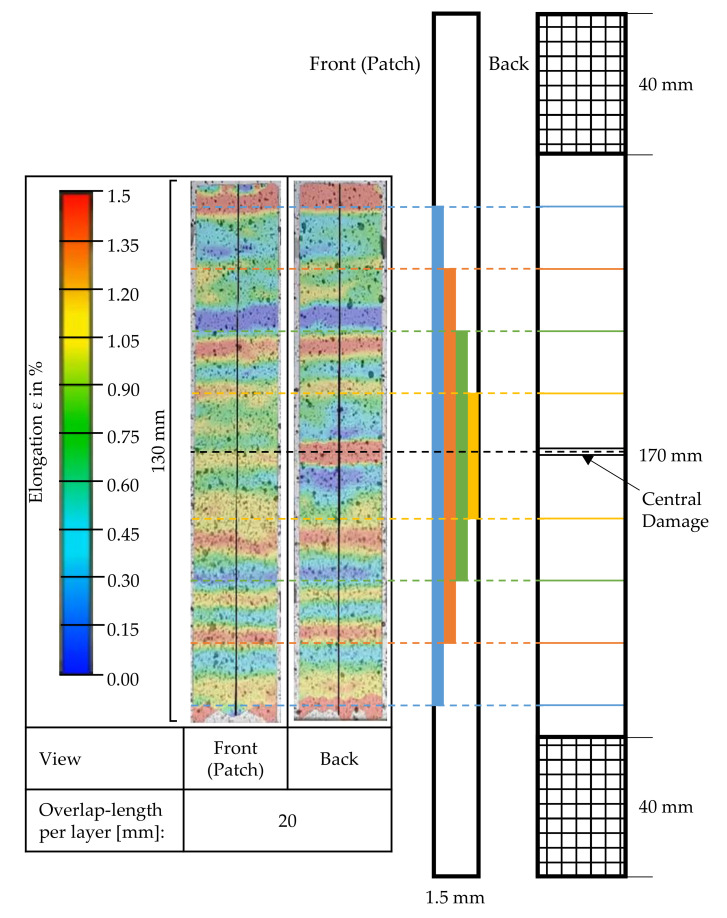
Results of the DIC during tensile testing just before the breakage of damaged samples ‘patch-repair’ with UD patch.

**Figure 10 materials-14-02711-f010:**
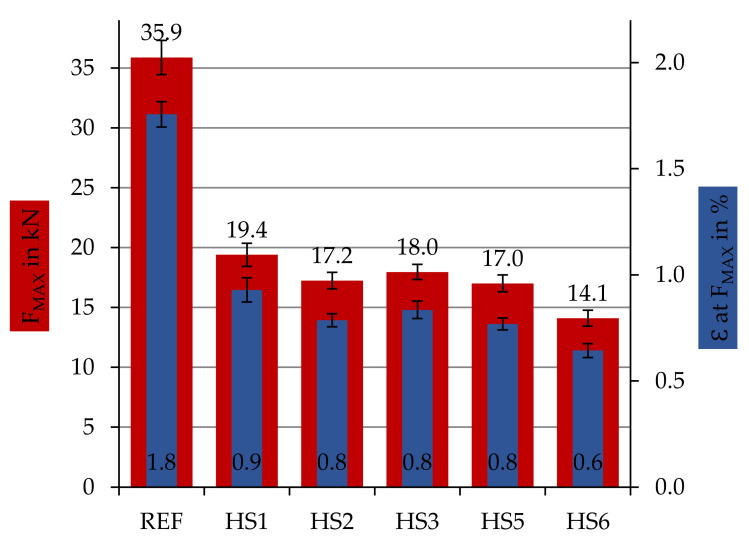
Specific tensile values of damaged samples repaired with UD patch material and different repair resin systems HA1-HS6 in comparison to the reference samples without any defect.

**Table 1 materials-14-02711-t001:** Repair resin systems.

Name	Resin System	Components	Gravimetric Ratio
HS1	Reference	RIMR135	100
		RIMH134	30
HS2	Reference + Additives 1	RIMR135	100
		RIMH134	30
		BYK-C 8013	5% of mass of matrix
		BYK-P 9920	1.5% of mass of matrix
HS3	Leuna-Resin 1	ER0051	100
		EC0052	25
HS5	Reference + Additives 2	RIMR135	100
		RIMH134	30
		BYK-C 8013	10% of mass of matrix
		BYK-P 9920	3% of mass of matrix
HS6	Leuna-Resin 1 + Additives	ER0051	100
		EC0052	30
		BYK-C 8013	5% of mass of matrix
		BYK-P 9920	1.5% of mass of matrix

## Data Availability

The data underlying this article will be shared on reasonable request from the corresponding author.

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
