# Peer review of "Novel Repair Procedure for CFRP Components Instead of EOL"

_materials, 2021, doi:10.3390/ma14112711_

Round 1
Reviewer 1 Report
In this paper the authors presented a study regarding CFRP plates’ reparation. In the first stage the authors deteriorate the matrix of CFRP and try to repair the used area by RTM or VARTM. In the second stage the reinforced material was repaired and the area was replaced by reinforced patch.
In order to improve the quality of the paper I recommend some specific changes.
Figures are too small. Please increase the size. It would be better if you put some patches details about staking sequence of the layers, from edges to center. Now, the figures are not representative, and at the same time are not clear. The figures must be representative for paper or section.
Row 95 ,pag. 3 ,, The suchlike stacked 95 patch-structure can be seen in Fehler! Verweisquelle konnte nicht gefunden wer-96 den. c) There is one parenthesis missing.
Here is better to introduce a reference. It is not clear what represents: ,,Fehler! Verweisquelle konnte nicht gefunden wer-96 den. c)”
Page4 rows 121-131. Here is not clear how you clean the damaged area of the matrix. I see that in the damaged area of resin (Figure 2) there is thermal degradation. How did you eliminate the resulting oxides of resin between layers of CF or between monofilaments? After degradation of the matrix there will be a contaminated area. The microstructure analyses of that area of CFRP and CF is necessary. Before and after RTM which we can see in Figure 2 In the right area marked with yellow, you have a brown area where the matrix is damaged or partial damaged. There must be more investigations or more details and explanations.
You mentioned several times that the matrix is dissolved from composite. It is necessary to give more details about this method. How did you dissolve a thermoset resin and how did you remove it from the CF layers or CF monofilaments in a way that the area must be free of contamination?
Pag. 5 rows 151-171 The explanation about CFRP repair procedure. Here is better to have a main scheme of the procedure. Where is the positioning of the metallic plates, where did you apply the vacuum bag, where did you apply the connection of the vacuum. You can insert some photos about the technology used.
And did you use RTM technology or VARTM because that point is not clear?
Figures 2 and 4 are in fact results and can be moved at results section.
It would be better if the authors use tensile strain results and diagrams. The force is not representative especially because the readers don’t know which is the thingness of the CFRP samples mentioned in the paper. The results can be compared to the other papers or research.
Page 9. Discussions. The results are predictable and the authors marked these things. As mentioned before the damaged of resin have a great influence in mechanical behavior of composite. The contaminated area is very difficult to clean end it’s hard to eliminate all the substances resulted from fibers. On the same time the affected-degraded matrix area is difficult to be evaluated. The authors mentioned that the samples were broken in the treated center area. It was interesting to see the microstructure in that area and to have some explanations about this remark.
Normally the CFRP is broken in many pieces and for the UD structure the broken area is specific (which we can see in many papers)
Reviewer 2 Report
This study presents a really interesting analysis about comparing different repair procuderes for CFRP components. The structire of the work is correct, the experimental procedure is perfectly described and the results and discussion are very easy to read and understand. The conclusions could be very usefull for researchers working in the field so I recommend the publocation of this work.
Not many changes are needed, because the paper is very detailed, and the information about the manufacturing process and the tensile process is significant.
I recommed to review the full texte because I found some typographical errors and some sentence written in german that should be corrected.
Please include the references related with the normatives.
It would be interesting to include some pictures showing the damaged material, to have a better idea about how relevant is the damage introduced by the milling. Is a throw-the-thickness damage? Is it only superficial? etc.
Finally, in Fig. 7 you can observe that the plies introduced by the patch repair, work as stress concentrators. I would like to know a little bit more about how is the breakage material in this case, because the strain distribution measure with DIC methodology seems to indicate that the material can be broken in different areas at the same time. What did you find in the material after tensile tests? New fibers were broken? Did the material present a fragile breakage? or was it detached?
Round 2
Reviewer 1 Report
Microstructure presented in fig.4 can be improved by a SEM images. Now the structures of CFRP are unrepresentative.
Author Response
Question 1: Microstructure presented in fig.4 can be improved by a SEM images. Now the structures of CFRP are unrepresentative.
Answer 1: No SEM images were taken, because macroscopically the patch is already visible. Therefore, the images from the light microscopy (see Fig. 8) were added during first review.